# Health Behaviours among Nursing Students in Poland during the COVID-19 Pandemic

**DOI:** 10.3390/nu14132638

**Published:** 2022-06-25

**Authors:** Ewa Kupcewicz, Kamila Rachubińska, Aleksandra Gaworska-Krzemińska, Anna Andruszkiewicz, Ilona Kuźmicz, Dorota Kozieł, Elżbieta Grochans

**Affiliations:** 1Department of Nursing, Collegium Medicum, University of Warmia and Mazury in Olsztyn, 14 C Zolnierska Street, 10-719 Olsztyn, Poland; 2Department of Nursing, Pomeranian Medical University in Szczecin, 48 Zolnierska Street, 71-210 Szczecin, Poland; kamila.rachubinska@pum.edu.pl (K.R.); grochans@pum.edu.pl (E.G.); 3Institute of Nursing and Midwifery, Medical University of Gdansk, 3a M. Sklodowskiej-Curie Street, 80-227 Gdansk, Poland; aleksandra.gaworska-krzeminska@gumed.edu.pl; 4Department of Basic Clinical Skills and Postgraduate Education for Nurses and Midwifes, Nicolaus Copernicus University in Torun, 1 Łukasiewicza Street, 85-821 Bydgoszcz, Poland; anna.andruszkiewicz@cm.umk.pl; 5Department of Internal Medicine and Community Nursing, Institute of Nursing and Midwifery, Faculty of Health Sciences, Medical College, Jagiellonian University, 12 Michalowskiego Street, 30-688 Krakow, Poland; ilona.kuzmicz@uj.edu.pl; 6Collegium Medicum, Jan Kochanowski University of Kielce, 19A IX Wieków Kielc Street, 25-369 Kielce, Poland; dkoziel@ujk.edu.pl

**Keywords:** pandemic COVID-19, health behaviours, satisfaction with life

## Abstract

(1) Background: An individual’s health status can be perceived as a consequence of their health behaviours. This research aimed to determine the intensity of health-promoting behaviours and to identify factors determining the health behaviours of nursing students during the COVID-19 pandemic. (2) Methods: This study included 894 nursing students from six universities in Poland, and it was conducted between 20 March and 15 December 2021. A diagnostic survey was applied as the research method, and the data were collected using the Health Behaviour Inventory and the Satisfaction With Life Scale. (3) Results: Nearly half of the students participating in the study (48.43%) declared that the intensity of their general health behaviours was low. A positive and significant correlation (r = 0.426) was found between general health behaviours and satisfaction with life. A regression model demonstrated general satisfaction with life to be a predictor of taking up health-related behaviours (18%; *β* = 0.34), as well as in terms of proper eating habits (4%; *β* = 0.15), prophylactic behaviours (6%; *β* = 0.21)*,* positive mental attitudes (26%; *β* = 0.44) and applied pro-health practices (10%; *β* = 0.25). (4) Conclusions: Most nursing students showed low levels of health-promoting behaviours. More research is needed on health behaviours and their determinants among nursing students, as it may be important in explaining the mechanisms of health behaviour formation.

## 1. Introduction

The World Health Organization (WHO) regards the COVID-19 pandemic as the greatest pandemic in modern times. Because of COVID-19, the governments of many countries imposed multiple restrictions to limit the risk of spreading the SARS-CoV-2 virus. Limiting people’s social activity is justified for health safety. Social isolation during the pandemic affected multiple spheres of human existence, and it impacted the functioning of individuals, including that of young people, within the mental, social and physical spheres [1,2,3,4]. The significant consequences of an increase in social distancing include those associated with the need to modify current behaviours; give up usual activities; and change the ways of eating, sleeping, recreation and leisure. However, several changes have taken place in the methods of communication and daily rhythms, and the Internet has become the main tool of work and study, as well as the place where interpersonal communication occurs during the COVID-19 pandemic [4,5,6,7].

Meanwhile, numerous studies have shown that both social isolation necessitated by the pandemic and excessive (often problematic) Internet use are associated with the intensification of many negative health behaviours and a significant deterioration in mental health status, especially among young people [7,8,9,10]. This is especially worrying because an individual’s lifestyle, i.e., behaviours manifesting a conscious attitude towards health and the responsibility for one’s and others’ health, is one of the major factors affecting one’s mental health. One’s behaviours are affected by multiple factors, both those associated with the environment (in social and cultural contexts) and those on the part of the individual, including specific convictions; expectations; motives; values; observations and subjectively perceived health status; and specific personality traits, including emotional states and satisfaction with life [11,12,13].

Satisfaction with life, described as a cognitive component of an individual’s subjective well-being, is one of the major indices of coping in life, as well as caring about one’s good mental and physical states. Satisfaction with life understood in this way can affect an individual’s health and longevity, making them take initiatives to boost their physical and mental well-being, i.e., taking up beneficial or anti-health behaviours [14,15]. Furthermore, it is demonstrated that individuals with a higher sense of satisfaction with life are more inclined to take on new initiatives. They are also more focused on self-development and setting new goals and challenges, including caring about their own mental and physical health [15,16]. This is confirmed by the findings of multiple studies conducted in many countries, which have shown that the COVID-19 pandemic has decreased the mental health status and satisfaction with life in the general population, as well as among students, and these levels correlated significantly with lifestyle-associated factors and pro-health behaviours, such as the level of physical activity, diet and the number of days spent at home [15,17,18]. In this context, it is extremely important to conduct scientific research concerning students’ health behaviours and changes during the pandemic, considering various factors and conditions, including their satisfaction with life.

The research aimed to determine the intensity of health-promoting behaviours and to identify factors determining the health behaviours of nursing students during the COVID-19 pandemic.

Given the stated aim, the following research questions were formulated:

What is the severity of health behaviours in general terms and regarding correct eating habits, preventive behaviours, positive mental attitudes and health practices among nursing students during the COVID-19 pandemic?

To what extent is health behaviour related to life satisfaction among nursing students during the COVID-19 pandemic?

To what extent do sociodemographic and lifestyle factors determine the health behaviour of nursing students during the COVID-19 pandemic?

## 2. Materials and Methods

### 2.1. Settings and Design

This study was conducted between 20 March and 15 December 2021 in six Polish universities with a nursing field of study. A total of 975 questionnaire sets were distributed among the nursing students, and 894 sets qualified for further analyses (91.69%). The collected data were encoded in an Excel spreadsheet, and a pooled analysis was performed. This study is part of a larger research project, which was approved (No. 3/2021) by the Senate Scientific Research Ethics Committee at the Olsztyn University in Olsztyn. The study met the criteria of a cross-sectional study [19].

### 2.2. Participants

Students who gave informed consent and were under 30 years old were enrolled in the study. Those who failed to give such consent were excluded from the study. After each dean’s consent was obtained, the researchers representing each university distributed the questionnaire forms among the students. The students were informed about the study objective and the method of completing the questionnaires, and they could ask questions and receive answers. They could withdraw from the study at any time. It took approximately 15 min to complete the questionnaire.

### 2.3. Research Instruments

A diagnostic survey method was applied, and two standardised research tools were used to collect the data. An original questionnaire developed by the authors was used to describe sociodemographic characteristics and selected lifestyle elements.

#### 2.3.1. Health Behaviour Inventory

The Health Behaviour Inventory developed by Z. Juczyński was used in this study to assess selected lifestyle indices. The general intensities of pro-health behaviour and four categories of health behaviours were evaluated:Proper eating habits (PNŻ), mainly taking into account the type of food eaten (e.g., wholemeal bread, fruit and vegetables);Prophylactic behaviours (ZP), which involve following health-related recommendations, and acquiring information on health and diseases;Health practices (PZ), which include daily habits related to sleep, recreation and physical activity;Positive mental attitudes (PNP), which include such psychological factors in the behaviours as avoiding too-strong emotions, stress, tension and depressing situations [20].

The IZZ scale contains 24 statements describing various health-related behaviours. The respondents identify the frequency of these health activities by assigning them points on a five-point scale, from “hardly ever”—1 point to “nearly always”—5 points. The points are summed up to provide the general health behaviour intensity index. The scores lie within an interval between 24 and 120 points. Higher scores indicate that more intense health behaviours were declared. After being converted to standardised units, the overall index is interpreted according to the properties of the sten scale. Scores between 1 and 4 sten are regarded as low, scores of 5 and 6 sten are regarded as average, and those from 7 to 10 sten are regarded as high. Moreover, the intensities of the four categories of health behaviours were calculated, with the index defined as the mean score in each category. The IZZ internal consistency based on Cronbach *alpha* was 0.85 for the whole scale, whereas it ranged from 0.60 to 0.65 for its four subscales. The test–retest examination gave a correlation coefficient of 0.88 [20].

#### 2.3.2. Satisfaction with Life Scale (SWLS)

The Satisfaction With Life Scale (SWLS) (developed by Ed. Diener et al., adapted for the Polish language by Z. Juczyński) is used to measure (hedonistic) mental well-being understood in terms of a conscious cognitive assessment of life. The scale contains five statements. Each statement is assigned points—from 1 (I definitely disagree) to 7 (I definitely agree). A respondent assesses the extent to which each of them applies to his/her life. The points are summed up, and the total score denotes the level of satisfaction with one’s life. The scores range from 5 to 35 points. The higher the score, the higher the sense of satisfaction with one’s life. After being converted to standardised units, the raw results are interpreted according to the properties of the sten scale. Scores between 1 and 4 sten are regarded as low, whereas scores from 7 to 10 sten are regarded as high. Scores of 5 and 6 sten are regarded as average. The SWLS internal consistency, based on the Cronbach *alpha*, was 0.81. The scale permanence index, determined in two tests six weeks apart, was 0.86 [20].

### 2.4. Statistical Analysis

A statistical analysis was conducted with STATISTICA v.13.3 (TIBCO, Palo Alto, CA, USA). The variables were described by descriptive statistics methods, with the following measures: arithmetic mean (M), median (ME), standard deviation (SD) and minimum–maximum (Min.–Max.). The confidence interval for the mean was also established (95% CI). The variable distributions were measured using the Kolmogorov–Smirnov test. The diverse impacts of sociodemographic variables and lifestyle-related variables on the general intensity of health behaviours and satisfaction with life were assessed using an intergroup one-way analysis of variance with the Fisher F test. Specific analyses were conducted using a post hoc test (LSD). The Pearson correlation (r) was used to examine the significance of the power of the correlation between the variables under analysis. A multiple regression analysis was conducted in order to determine health behaviour predictors. The interpretation of the correlation power between the analysed variables was based on Guilford’s classification: |r| = 0—no correlation, 0.0 < |r| ≤ 0.1—slight correlation, 0.1 < |r| ≤ 0.3—weak correlation, 0.3 < |r| ≤ 0.5—average correlation, 0.5 < |r| ≤ 0.7—high correlation, 0.7 < |r| ≤ 0.9—very high correlation, 0.9 < |r| < 1.0—nearly full correlation and |r| = 1—full correlation [21]. The level of significance of *p* < 0.5 was adopted.

## 3. Results

The study included 894 nursing students, comprising 822 females (91.95%) and 72 males (8.05%). The mean participant age was 20.73 years (SD = 1.81) (Table 1).

A statistical analysis of the collected data was performed, and the mean values of the variables were calculated for the whole group of Polish nursing students.

The overall health behaviour score in the study group was determined to be 77.75 points (SD = 13.52) on a scale from 24 to 120. The following mean values were calculated for the individual health behaviour categories: prophylactic behaviours 3.41 (SD = 0.72), positive mental attitudes 3.24 (±0.72), proper eating habits 3.21 (SD = 0.81) and health practices 3.10 (SD = 0.73) (Table A1).

Further analyses involved converting the general health behaviour index and Satisfaction With Life Scale into standardised units, which were interpreted according to the properties of the sten scale. The results for nearly half of the nursing students (48.43%) ranged from 1 to 4 sten, which was indicative of a low health behaviour intensity in the general perspective. Results between 7 and 10 sten, indicating a high general health behaviour intensity, were noted for merely 14.77% of the participants. A slightly different case was presented in interpreting the results that characterised satisfaction with life among the students. Results between 1 and 4 sten, indicative of a low sense of satisfaction with life, were noted for 38.03% of the respondents, while those ranging between 7 and 10 sten, regarded as high, were calculated for 25.39% of them (Figure 1).

The next step involved verifying whether the general intensity of health behaviours is associated with satisfaction with life among the nursing students participating in the study. A significant correlative relationship at an average level (r = 0.426; *p* < 0.0001) was noted between the general intensity of health behaviours and satisfaction with life. This means that the variables under study increase in a mutually proportional manner. In Figure 2, a scatter plot diagram shows a graphic interpretation of the Pearson correlation coefficient (r).

The diverse impacts of sociodemographic variables and lifestyle-related variables on the general intensity of health behaviours and satisfaction with life were assessed in further analyses. A one-way intergroup analysis of variance revealed differences in the general intensity of health behaviours between the first-, second- and third-year students (F = 5.97; *p* < 0.002). Detailed analyses showed that the general health behaviour intensity index of the first-year students was significantly lower than that of the second-year (*p* < 0.001) and third-year students (*p* < 0.001). Further analyses showed that a significant role in health behaviour intensity was played by the number of meals consumed per day (F = 27.76; *p* < 0.0001). Young people who consumed several meals a day declared higher levels of satisfaction with life (Table 2). The factors that had a significant impact on satisfaction with life among the nursing students in the second year of the COVID-19 pandemic included four variables, with reduced social contacts and a subjective health status assessment being among them. Differences were demonstrated in satisfaction with life between students who reduced their social contacts to large, average and small extents (F = 4.24; *p* < 0.005). Detailed analyses showed that satisfaction with life in students who reduced their social contacts to a large extent was significantly lower than in those who did it to an average (*p* < 0.05) or a small extent (*p* < 0.01). The subjective health status assessment significantly impacted the students’ satisfaction with life (F = 33.03; *p* < 0.0001) in the study group. Detailed analyses showed that satisfaction with life in the students who saw their health status during the pandemic as poor was significantly lower than in those who saw it as good (*p* < 0.01) and very good (*p* < 0.001) (Table 2).

The next step in the statistical analyses involved identifying health behaviour predictors from among all the sociodemographic and lifestyle-related variables considered in the study.

An analysis of the results shown in Table 3 revealed that satisfaction with life proved to be the main predictor of the general intensity of health behaviours with the greatest predictive power (18%). This regression model explained 27% of the variability of the results. The regression coefficient for the main predictor was positive (ßeta = 0.34; R^2^ = 0.27), which is indicative of a positive correlation. This means that the higher the satisfaction with life demonstrated by the nursing students, the higher the value assigned to the general intensity of health behaviours.

Further analyses involved determining predictors for individual categories of health behaviours. The analysed independent variables were introduced to the regression equation each time. Satisfaction with life proved to be the main predictor in each health behaviour category. However, it had the greatest predictive power (26%) for the health behaviour category referred to as positive mental attitudes, including such psychological factors as avoiding too-strong emotions, stress and tension. The regression coefficient was positive (ßeta = 0.44; R^2^ = 0.32), which is indicative of a positive correlation. The second variable in this health behaviour category—subjective health status assessment—explained only 3% of the variability of the results. The other two variables (number of meals and study year) demonstrated only a minor predictive power (total 3%).

The analysis revealed that satisfaction with life had the predictive power of 10% in the health behaviour category referred to as pro-health practices, including everyday habits associated with sleep, recreation and physical activity. The regression coefficient was positive (ßeta = 0.25; R^2^ = 0.15), which is indicative of a positive correlation. The other two variables (number of meals and subjective health status assessment) explained 5% of the variability of the results.

Satisfaction with life regarding proper eating habits, which mainly included the type of food eaten (e.g., wholemeal bread, fruit and vegetables), had a predictive power of 4%. The regression coefficient was positive (ßeta = 0.15; R^2^ = 0.10), which is indicative of a positive correlation. This regression model explained 10% of the variability of the results.

The data show that satisfaction with life had a predictive power of 6% in the health behaviour category referred to as prophylactic behaviours, associated with following health recommendations and acquiring information on health and diseases. The regression coefficient was positive (ßeta = 0.21; R^2^ = 0.10), which is indicative of a positive correlation. Furthermore, the second variable (the number of meals) explained 2% of the variability of the results. The other three variables (subjective health status assessment, year of studies and age) did not play a significant role in predicting health behaviours in the “prophylactic behaviours” category (Table 3).

## 4. Discussion

Numerous multi-centre studies have indicated a high risk of negative health behaviours during the COVID-19 pandemic and the need to identify and monitor them [5,22,23,24].

This study’s findings indicate that nearly half of the respondents (48.43%) had scores indicating a low intensity of general health behaviours, which may provide grounds for further studies on the factors that stimulate individuals to take up health behaviours (so-called “health-related motives”). Students from Spain and Slovakia participating in a different study on health behaviours had a higher percentage of results at a higher level than students from Poland [25]. When comparing the current results with the data gathered by other authors on health behaviours, a certain differentiation can be observed. The tendency of a dominant average level of health-related behaviours before the COVID-19 pandemic was proven among Polish students from Poznań universities [26], academic youth studying at Wszechnica Świętokrzyska [27] and Lublin medical students [28].

Radosz et al. studied health behaviours among students in three fields of medical studies, and they observed considerably lower results concerning the intensity of general health behaviours among students of physiotherapy than among students of midwifery and nursing [29]. All of this shows that the measurement of health behaviours among Polish students is a complex issue that requires in-depth analyses.

Badura-Brzoza et al. surveyed medical personnel during the first wave of the COVID-19 pandemic and recorded a slightly higher general health behaviour score than that recorded in this study. They also found that the scores for general health behaviours were significantly higher among nurses than among doctors [30]. Many researchers have pointed out that members of healthcare personnel experience the negative effects of the COVID-19 pandemic on various levels, with mental health being one of them. An intensification of mental health issues, among both healthcare professionals and the general population, is associated with depression, anxiety, insomnia and PTSD [31,32,33]. Villadsen et al. demonstrated (based on four domestic longitudinal cohort studies in the UK) that mental health deterioration in a group of 10,666 participants was associated with detrimental health behaviours—changes in diet, physical activity and sleep quality [34]. Kim et al. also surveyed nursing students and found that age, health status, knowledge and risk perception significantly affected preventive health behaviour, which was found to be positively correlated with knowledge and risk perception. An educational program, which considers student age, health level, knowledge and perception of risk, is required to enhance the preventive health behaviour of nursing students in view of COVID-2019 [35].

The analysis conducted in this study of health behaviour intensities in different categories revealed that the category with the highest score was the “prophylactic behaviours” category, which involves following health recommendations and acquiring information on health and diseases. One can conclude that this is associated with the study programme. Pro-health practices, including daily habits associated with sleep, recreation and physical activity, received the lowest scores from nursing students. This shows that the habitual health-related activities of Polish nursing students should be particularly emphasised.

In other studies, students took up activities in the “pro-health practices” and “positive mental attitudes” categories more frequently than in the categories of proper eating habits and prophylactic behaviours [36,37].

Interesting findings indicative of significant differences in preferred health behaviours among nurses and doctors in the “positive mental attitudes” and “prophylactic behaviours” categories were also presented by Badura-Brzoza et al., who found that higher scores were noted for nurses [30].

Satisfaction with life is a factor that affects an individual’s attitude towards health [11,12,13,36]. Meanwhile, a reduction in numerous everyday activities, including the pandemic-related social isolation, significantly impacted psychosocial health and satisfaction with life in individuals in all age groups [38,39].

Many studies have confirmed the negative impact of the COVID-19 pandemic on people’s satisfaction with life [15,40,41,42,43]. Some researchers also found that pandemic-related social relation disorders were predictors of lower satisfaction with life [15]. However, higher satisfaction with life positively impacted the achievement of a better quality of life, alleviation of the negative effects of stressful events and coping with difficult situations [44].

Every fourth student had a high satisfaction with life in this study. A low satisfaction with life was more typical for the respondents who strongly reduced their social contacts and those who saw their health status as poor. As expected, the overall health behaviour index was positively correlated with satisfaction with life. Interesting findings on satisfaction with life were reported by researchers in a multi-centre study conducted during the first wave of the COVID-19 pandemic among students in nine countries, namely, Czechia, Poland, Slovenia, Germany, Ukraine, Russia, Turkey, Columbia and Israel. The majority of the students (60.54%) were found to be satisfied with their lives, whereas a low self-assessment of their health status was a predictor of low satisfaction with life, regardless of the country [15]. Notable studies on the subject include a study conducted by Gultekin et al. of a group of 336 students who studied at nine faculties of the Dokuz Eylul University in 2019–2020. The researchers demonstrated that students’ satisfaction with life decreased with increasingly risky health behaviours [42].

A different study, conducted by Machul et al., found that satisfaction with life among foreign and Polish medical students tended to be similar. However, there were differences between students of the two genders. The SWLS scores noted for Polish female students were higher than those noted for foreign students [45]. A study conducted by Pan et al. among Chinese students in Australia revealed that foreign female students showed higher satisfaction with life than male students [46]. The health behaviours and satisfaction with life as examined in this study were not affected by gender. It seems that the programme, the field of study and the year of study play a significant role in popularising health behaviours [29,44]. This was also confirmed in the current study: first-year students had lower scores for general health behaviours than students in later years. The regression analysis showed satisfaction with life to be the main predictor of health behaviours, with the greatest predictive power identified for the “positive mental attitudes” category. Therefore, satisfaction with life can be regarded as an adequate measure of one’s mental and physical states, and health status is reflected in subjective well-being, which includes emotional, feeling-related and cognitive elements [20]. It is evident from this study that satisfaction with life is an important human health resource, conducive to active care for health. In the current research, 25.39% of nursing students indicated a high sense of satisfaction with life, 36.58% indicated an average satisfaction with life and 38.03% indicated a low satisfaction with life. A statistically significant correlation was also found in the average level between life satisfaction and the health behaviours of the respondents. Relationships between life satisfaction and some behavioural determinants of health (mainly nutritional) have been confirmed in Chilean research [47]. However, a positive relationship between the level of life satisfaction and the undertaking of physical activity was noted in American [48,49], Australian [50] and Croatian [51] studies [36].

### Limitations and Implications Regarding Professional Practice

This study of the health-related behaviours of nursing students is one of the first multi-centre studies conducted during the COVID-19 pandemic in Polish universities. Its findings can be useful in planning prophylactic measures and adapting intervention strategies aimed at applying student-motivation-supporting techniques targeted at pro-health behaviours. Health practices are an area requiring special measures. It is worth considering the possibility of providing institutional support and psychological aid to students, particularly in situations when, for example, negative eating habits are linked to mental health or a lack of physical activity results from feeling too much anxiety. There are plans to include these issues in future studies. The authors point out that there are certain limitations to this study, which are associated with the lack of data on the intensity of health behaviours among nursing students just before the outbreak of the COVID-19 pandemic; these data can be important for a comparison of the variables under analysis. Students who had already been diagnosed before the study with, for example, mood or eating disorders and those who had family problems not related to studying were also not excluded.

## 5. Conclusions

The majority of the nursing students surveyed showed generally low levels of health-promoting behaviours. In individual categories, there were variations in the intensities of the declared health behaviours. The highest intensity was recorded in the category of preventive behaviours, while the lowest intensity was recorded in the category of health practices.

Greater intensities of declared health behaviours are related to greater satisfaction with life among nursing students during the COVID-19 pandemic.

A student’s year of study, the number of meals consumed per day and a subjective assessment of one’s health turned out to be the determining factors of health behaviours during the COVID-19 pandemic.

Life satisfaction, considered a valid measure of well-being, assumed the role of a predictive factor in undertaking health behaviours in general, as well as in terms of correct eating habits, preventive behaviours, positive mental attitudes and health practices among Polish nursing students during the COVID-19 pandemic.

Preventive activities should be carried out systematically in universities in relation to nursing students who, in the future, will be promoters of a healthy lifestyle. More research is needed on health behaviours and their determinants among nursing students, as it may be important in explaining the mechanisms of health behaviour formation.

## Figures and Tables

**Figure 1 nutrients-14-02638-f001:**
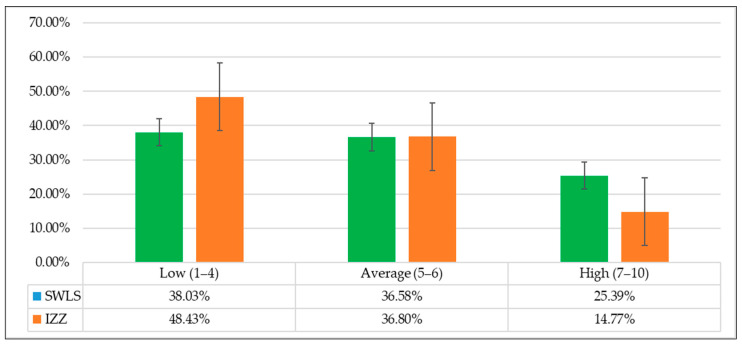
Result distribution for a sense of satisfaction with life and health behaviours on the sten scale. Abbreviations: SWLS—Satisfaction With Life Scale, IZZ—Health Behaviour Inventory.

**Figure 2 nutrients-14-02638-f002:**
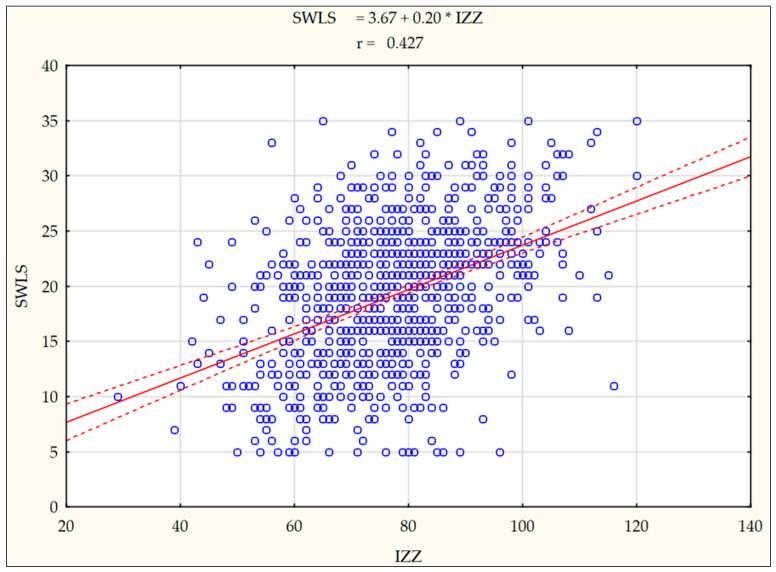
The character and intensity of the correlation between satisfaction with life and health behaviours from a general perspective among the study participants—Pearson correlation coefficients (r). Abbreviations: SWLS—Satisfaction With Life Scale, IZZ—Health Behaviour Inventory, Solid red line—straight line of the linear regression SWLS = 3.67 + 0.20 * IZZ, Dotted red line—line of the regression strip with a confidence interval at 0.95.

**Table 1 nutrients-14-02638-t001:** Sociodemographic characteristics of the study group.

Variables		Total N = 894	
		Number	%
University/College name	Pomeranian Medical University in Szczecin	215	24.05
	University of Warmia and Mazury in Olsztyn	175	19.57
	Medical University of Gdańsk	143	16.00
	Nicolaus Copernicus University in Toruń, the Collegium Medicum in Bydgoszcz	171	19.24
	Jagiellonian University in Kraków	132	14.77
	Jan Kochanowski University of Kielce	57	6.38
Gender	female	822	91.95
	male	72	8.05
Study year	first	397	44.41
	second	289	32.33
	third	208	23.27
Age (years)	≤20	481	53.80
	21–22	319	35.68
	≥23	94	10.51
Place and form of residence	with family/someone close	621	69.46
	on their own	273	30.54
Number of hours spent working on a computer	≤5	433	48.43
	6–9	302	33.78
	≥10	159	17.79
Number of consumed meals per day	1–2	104	11.63
	3	382	42.73
	4	280	31.32
	≥5	128	14.32
Restriction of physical activity during the pandemic	no	211	23.60
	yes, to a small extent	161	18.01
	yes, to a medium extent	278	31.10
	yes, to a considerable extent	244	27.29
Subjective health status assessment during the pandemic	bad	24	2.68
	good/average	613	68.57
	very good	257	28.75
Restriction of social contacts during the pandemic	very high	141	15.77
	considerable	360	40.27
	medium/average	229	25.62
	to a small extent	164	18.34

Abbreviations: N—number of subjects.

**Table 2 nutrients-14-02638-t002:** The significance of the impact of sociodemographic and lifestyle-related variables on the general intensity of health behaviours and satisfaction with life.

Variables		N = 894 (%)	IZZ					SWLS				
			M	SD	F	*p*-Value	Post Hoc (NIR)	M	SD	F	*p*-Value	Post Hoc (NIR)
Gender	female	822 (91.95)	77.98	13.52	2.97	0.09		19.33	6.38	0.78	0.37	
	male	72 (8.05)	75.13	13.31				18.64	6.04			
Study year	first (A)	397 (44.41)	76.05	13.16	5.97	0.002	A < B ** A < C **	18.90	6.32	1.5	0.22	
	second (B)	289 (32.33)	78.80	13.49				19.39	6.49			
	third (C)	208 (23.27)	79.57	13.90				19.82	6.21			
Age (years) M = 20.73; SD = 1.81	≤20	481 (53.80)	77.27	13.11	1.01	0.36		19.26	6.17	0.77	0.45	
	21–22	319 (35.68)	78.02	13.97				19.08	6.61			
	≥23	94 (10.51)	79.33	14.03				20.01	6.43			
Place and form of residence	with family/someone close	621 (69.46)	78.02	13.30	0.76	0.38		19.38	6.40	0.6	0.43	
	on their own	273 (30.54)	77.16	14.01				19.03	6.26			
Number of hours spent working on a computer M = 6.08; SD = 3.19	≤5 (A)	433 (48.43)	78.59	13.77	1.84	0.15		19.96	6.14	5.4	0.004	A < B **
	6–9 (B)	302 (33.78)	77.27	13.19				18.43	6.47			
	≥10 (C)	159 (17.79)	76.38	13.36				19.00	6.54			
Number of meals M = 3.48; SD = 0.87	1–2 (A)	104 (11.63)	69.03	14.19	27.76	0.0001	A < B,C,D *** B < C,D *** C < D ***	16.22	6.81	12.17	0.0001	A < B,C,D *** B < C ** B < D *
	3 (B)	382 (42.73)	76.55	13.35				19.01	6.37			
	4 (C)	280 (31.32)	80.13	12.32				20.31	5.86			
	≥5 (D)	128 (14.32)	83.25	12.00				20.27	6.17			
Restriction of physical activity during the pandemic	no	211 (23.60)	78.51	13.65	3.46	0.02		19.70	6.29	1.79	0.14	
	yes, to a small extent	161 (18.01)	79.11	13.64				19.55	6.25			
	yes, to a medium extent	278 (31.10)	78.45	12.69				19.49	6.53			
	yes, to a considerable extent	244 (27.29)	75.41	14.03				18.48	6.25			
Subjective health status assessment during the pandemic	bad (A)	24 (2.68)	66.08	11.45	44.02	0.0001	A < B ** A < C *** B < C ***	14.71	5.99	33.03	0.0001	A < B ** A < C *** B < C ***
	good/average (B)	613 (68.57)	75.74	13.03				18.42	6.14			
	very good (C)	257 (28.75)	83.65	12.74				21.73	6.15			
Restriction of social contacts during the pandemic	very high (A)	141 (15.77)	77.38	14.35	0.88	0.44		18.21	5.98	4.24	0.005	A < C * A < D **
	considerable (B)	360 (40.27)	77.80	13.64				18.89	6.42			
	medium/average (C)	229 (25.62)	78.76	13.01				19.62	6.26			
	to a small extent (D)	164 (18.34)	76.57	13.22				20.55	6.47			

Abbreviations: IZZ—Health Behaviour Inventory, SWLS—Satisfaction With Life Scale, M—arithmetic mean, SD—standard deviation, F—Fisher test, NIR—post hoc test (Smallest Significant Difference Test). Statistically significant: * *p* < 0.05; ** *p* < 0.01; *** *p* < 0.001.

**Table 3 nutrients-14-02638-t003:** Health behaviour predictors.

Variables		R^2^	ßeta	ß	t	*p*-Value
IZZ	Constant value			44.80	15.24	0.001
	SWLS	0.18	0.34	0.73	11.38	0.001
	Number of meals	0.23	0.19	2.94	6.49	0.001
	Subjective health status assessment during the pandemic	0.26	0.19	5.09	6.24	0.001
	Study year	0.26	0.12	2.05	2.65	0.008
	Restriction of social contacts during the pandemic	0.27	−0.08	−1.06	−2.53	0.01
	R = 0.52; R^2^ = 0.27; corrected R^2^ = 0.27					
PNŻ	Constant value			13.10	9.30	0.001
	SWLS	0.04	0.15	0.12	4.57	0.001
	Number of meals	0.06	0.14	0.76	4.17	0.001
	Subjective health status assessment during the pandemic	0.08	0.14	1.35	4.15	0.001
	Restriction of social contacts during the pandemic	0.08	−0.09	−0.45	−2.69	0.007
	Restriction of physical activity during the pandemic	0.10	−0.09	−0.37	−2.58	0.01
	R = 0.33; R^2^ = 0.10; corrected R^2^ = 0.10					
ZP	Constant value			3.04	8.57	0.001
	SWLS	0.06	0.21	0.02	6.10	0.001
	Number of meals	0.08	0.12	0.10	3.80	0.001
	Subjective health status assessment during the pandemic	0.09	0.11	0.16	3.26	0.001
	Study year	0.10	0.12	0.11	2.69	0.007
	Age	0.10	−0.10	−0.04	−2.22	0.03
	R = 0.32; R^2^ = 0.11; corrected R^2^ = 0.10					
PNP	Constant value			6.90	8.57	0.001
	SWLS	0.26	0.44	0.31	15.07	0.001
	Subjective health status assessment during the pandemic	0.29	0.18	1.58	6.18	0.001
	Number of meals	0.32	0.15	0.76	5.30	0.001
	Study year	0.32	0.08	0.43	2.78	0.005
	R = 0.57; R^2^ = 0.33; corrected R^2^ = 0.32					
PZ	Constant value			11.42	10.13	0.001
	SWLS	0.10	0.25	0.17	7.64	0.001
	Number of meals	0.13	0.17	0.85	5.40	0.001
	Subjective health status assessment during the pandemic	0.15	0.13	1.18	4.11	0.001
	R = 0.39; R^2^ = 0.153; corrected R^2^ = 0.15					

Statistically significant: *p* < 0.05; *p* < 0.01; *p* < 0.001. Abbreviations: IZZ—Health Behaviour Inventory, PNŻ—proper eating habits; ZP—prophylactic behaviours; PNP—positive mental attitudes; PZ—health practices, SWLS—Satisfaction With Life Scale.

## Data Availability

The data presented in this study are available on request from the first author.

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
