# Peer review of "Health Behaviours among Nursing Students in Poland during the COVID-19 Pandemic"

_nutrients, 2022, doi:10.3390/nu14132638_

Round 1
Reviewer 1 Report
The manuscript called "Health behaviours with among nursing students in Poland during the COVID-19 pandemic" is an interesting piece using a large sample of 894 nursing students from six universities in Poland. The paper is consider very attractive and actual for the readers. The abstract and introduction are well written and just a minor proofreader is required for a final review.
Regarding to the methodology is clear the study and carefully controlled. Nonetheless, suggests just to clarify how the data are saved and how was organized the data protection. In the results no modifications are required. The statistics are well presented and organized in the manuscript.
However, in the discussion, requires to explore more the results analysed. In the sentence: "This was also confirmed in the current study: first-year students had lower scores for general health behaviours than those of later years. The regression analysis showed satisfaction with life to be the main predictor of health behaviours, with the greatest predictive power identified for the “positive mental attitude” category. Therefore, satisfaction with life can be regarded as an adequate measure of one’s mental and physical state, and health status is reflected in subjective well-being, which includes emotional, feelings-related and cognitive elements" - is required to explore in few more words the findings from the study. Another suggestion is to explore graphically the sample with more details about general characteristics, if other data are available for complementary information.
The conclusion is clear but can contain more recommendations for future studies. This can helps future investigation with similar sample. Minor revision is required but the paper is really interesting.
Author Response
Dear Reviewers, Thank you very much for a thorough editorial assessment of my manuscript, positive opinions, as well as the reviewers’ remarks. I used them as an important guide to improving the quality of my paper. The implemented corrections were done strictly according to their comments. All changes made in the text are marked in red and yellow. I have enclosed the re-edited manuscript and cover letter as responses to Reviewers, detailing how I followed their suggestions. Thank you very much for your kind consideration of my paper. Yours sincerely, Ewa Kupcewicz, PhD

Reviewer 2 Report
Thank you for giving me the opportunity of reviewing this paper that approaches an interesting subject. During university life, students should consolidate their healthy habits, which will be reflected later in their adult life. This is especially important in the case of students who will become health professionals. However, much of the literature concludes that students report bad habits during the university period, so it is essential to detect the factors that predispose them to unhealthy behavior. Therefore, this manuscript presents several weaknesses.
1. Introduction:
In the second paragraph, the authors focus on describing how covid-19 has affected the lifestyle habits of young people, but the study population is composed of students. They should refer to any study conducted with university students.
2. Objective:
Correlation should be substituted by association. Also, the second part of the objective “to seek predictors of the analyzed health behaviors” is not well understood. It should be rewritten.
3. Methodology:
The design of the study must be indicated.
2.2. Participants: Could the authors include the response rate? Have the authors made a predetermination of the sample size to ensure that the number of respondents is sufficient so as not to compromise the validity of the results.
2.3.1. Health behavior inventory: “The general intensity of pro-health behavior” is later named Overall Health Behaviour score in results and O/ZZ- health behavior inventory (general) in table 1. Differences in terminology make interpretation difficult.
4. Results:
Name of Subsection 3.1. “Analysis of variables” is not very accurate, as variables are analyzed in all sections.
“The Kolmogorov- Smirnov test results show that the variable distribution does not differ significantly in most cases from the normal distribution” is not usually appointed in the beginning of the section “results”. In addition, it is not very accurate. Which variables follow a normal distribution? can be shown more visually in Table 1.
Table 1. SD usually goes beside the average.
Figure 1. it is recommended to keep the same order as in the table above (O/ZZ, SWLS). Is there any difference statistically significant?
Figure 2. “korelacja “is not in English. Pearson correlation coefficient (not plural).
Table 2. Please, change the word “explanations”. In this table Health behavior inventory (general) is defined as IZZ, which is different from the previous table and the methodology.
Are the terms beta and b abbreviations of standardized coefficient and non- standardized coefficient? It is not well understood. Where do the authors observe that number of meals have a predictive power of 5% in table 3?. All the predictive powers are observed in table 3?
Table 3. p-value: 0.000 it incorrect. Authors should indicate p<0,001. In PZ there is not the name of the constant value. * should not be indicated in the p-value.
Discussion:
The discussion is not well structured and there is little interpretation of the results obtained. It should be enriched by contrasting them with more evidence.
Author Response

(The authors gave the same response as above.)

Round 2
Reviewer 2 Report
After analyzing the new manuscript provided, I believe that the authors have made substantial improvements so I believe it can be published in its current form.
Author Response
Thank you very much for your comments.